# 'More than just numbers on a page?' A qualitative exploration of the use of data collection and feedback in youth mental health services

Craig Hamilton[1]*, Kate Filia[1,2], Sian Lloyd[1], Sophie Prober[1], Eilidh Duncan[3]

**1** Orygen, Melbourne, Australia, **2** Centre for Youth Mental Health, University of Melbourne, Melbourne, Australia, **3** Health Services Research Unit, Institute of Applied Health Sciences, University of Aberdeen, Aberdeen, United Kingdom

\* craig.hamilton@orygen.org.au

**Data Availability Statement:** All relevant data are within the paper and its Supporting information files.

## Abstract

### Objectives

This study aimed to explore current data collection and feedback practice, in the form of monitoring and evaluation, among youth mental health (YMH) services and healthcare commissioners; and to identify barriers and enablers to this practice.

### Design

Qualitative semi-structured interviews were conducted via Zoom videoconferencing software. Data collection and analysis were informed by the Theoretical Domains Framework (TDF). Data were deductively coded to the 14 domains of the TDF and inductively coded to generate belief statements.

### Setting

Healthcare commissioning organisations and YMH services in Australia.

### Participants

Twenty staff from healthcare commissioning organisations and twenty staff from YMH services.

### Results

The umbrella behaviour 'monitoring and evaluation' (ME) can be sub-divided into 10 specific sub-behaviours (e.g. planning and preparing, providing technical assistance, reviewing and interpreting data) performed by healthcare commissioners and YMH services. One hundred belief statements relating to individual, social, or environmental barriers and enablers were generated. Both participant groups articulated a desire to improve the use of ME for quality improvement and had particular interest in understanding the experiences of young people and families. Identified enablers included services and commissioners working in

**Funding:** The authors received no specific funding for this work.

**Competing interests:** The authors have declared that no competing interests exist.

partnership, data literacy (including the ability to set appropriate performance indicators), relational skills, and provision of meaningful feedback. Barriers included data that did not adequately depict service performance, problems with data processes and tools, and the significant burden that data collection places on YMH services with the limited resources they have to do it.

## Conclusions

Importantly, this study illustrated that the use of ME could be improved. YMH services, healthcare commissioners should collaborate on ME plans and meaningfully involve young people and families where possible. Targets, performance indicators, and outcome measures should explicitly link to YMH service quality improvement; and ME plans should include qualitative data. Streamlined data collection processes will reduce unnecessary burden, and YMH services should have the capability to interrogate their own data and generate reports. Healthcare commissioners should also ensure that they provide meaningful feedback to their commissioned services, and local and national organisations collecting youth mental health data should facilitate the sharing of this data. The results of the study should be used to design theory-informed strategies to improve ME use.

## Introduction

The collection, analysis, and feedback of health services data plays an essential role in the improvement of health care [1–5]. Globally, shortcomings in the quality of mental health care have been identified and there is substantial interest in enhancing the use of data to address these. Opportunities for this include strategies designed to bring about changes in healthcare provider behaviour such as routine outcome measurement [6]; audit and feedback [5]; and monitoring and evaluation [2,7–13]. These strategies can improve care and patient outcomes but the effects are highly variable [5] and their potential has not been fully realised [14]. Knowing more about the conditions under which collection and feedback of data works to change practice, and identifying the barriers to its effective use, helps us to understand how to optimise it [15]. There is a recognised risk within the healthcare improvement field that the "effort invested in collecting information (which is essential) is not matched by effort in making improvement" [16].

This paper focuses on the use of monitoring and evaluation (ME) in the context of Australian youth mental health care. ME involves the systematic collection and analysis of program data (e.g. program activity, patient outcomes) to provide strategic information, which can be used for decision-making by program managers and healthcare commissioners. Monitoring is a continuous process which tracks progress in implementation and performance, often against indicators and targets [17]. Evaluation is a periodic activity, which can identify the extent to which intended objectives have been achieved and can provide insight into what has contributed to their achievement or non-achievement [17].

### Youth mental health care in Australia

In response to the high burden and incidence of mental ill-health among young people, and inadequacies of the mental health system to meet their needs, numerous countries have developed and implemented youth mental health (YMH) services targeted to young people aged 12 to 25 years [18–22]. In Australia, YMH services are typically commissioned by 31 Primary

Health Networks (PHNs) [23] and delivered by local or national non-government organisations. A significant proportion of services operate as part of a national franchise led by the headspace National Youth Mental Health Foundation (110 centres by 2019) [24,25].

There is recognition of the importance of the collection and feedback of data within mental health services [2,6,7,9,26], and the practice of ME is perceived as an integral component of the commissioning process and contemporary YMH service provision [18,24,27–29]. Despite this, healthcare commissioners report that they find it challenging to make meaningful use of ME data collected from YMH services [30]. Little is known about how YMH services and healthcare commissioners currently use ME and given its potential to contribute to the improvement of mental health care provided to young people, it is essential that we understand what helps and hinders its use. This study aimed to explore current ME practice among YMH services and healthcare commissioners in Australia, and to identify individual and environmental barriers and enablers to these practices.

## Methods

### Sampling and recruitment

Participants were purposively sampled to ensure representation across healthcare commissioning organisations and YMH services, from a variety of roles/responsibilities, and with good coverage of geographical areas within Australia.

The research team, members of which were employed at Orygen, a national youth mental health organisation [31], sent an email invitation to an existing network of contacts working in healthcare commissioning organisations and YMH services (n = 240). Snowball sampling was also used, where recipients of the original email invitation were asked to forward the email to their contacts they deemed relevant (based on the information provided in the email). A flow-chart detailing participant recruitment to the two sample groups is provided in Fig 1.

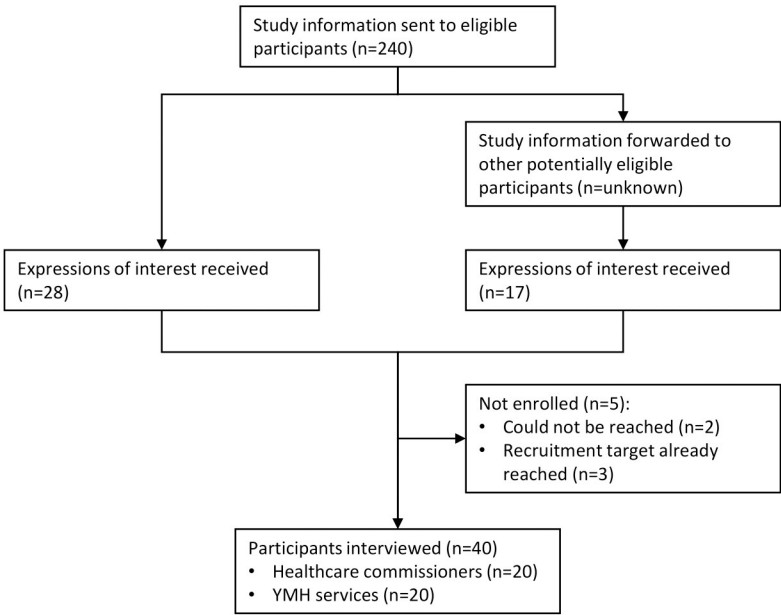

**Fig 1. Flow chart of participants' recruitment to the study.**

## Interviews

Semi-structured interviews were conducted by the lead author (CH), a male completing a Master of Public Health, and supervision was provided by a senior researcher (ED). Data collection and analyses were informed by the Theoretical Domains Framework (TDF) [32]. The TDF incorporates 33 theories of behaviour change and is used to explore and identify factors which inhibit or enable professional behaviour change [32–34]. CH and ED met frequently, with expertise in TDF drawn from ED where required, related to reviewing the topic guide; coding guidelines; interview recordings; and all coding. While an interview topic guide informed by the TDF [32] was developed, the researcher encouraged a natural flow to the interviews; as such, they were semi-structured depending on when and how topics were raised by the participant. The topic guide was piloted with two mental health professionals with experience of managing YMH services and amended to improve clarity and reduce length. The topic guide is provided in S1 Appendix.

Interviews were conducted between June 2020 and August 2020 using Zoom videoconferencing software, apart from one telephone interview. Due to the COVID-19 pandemic, most participants were at home during their interview but a small minority were in a private office at their usual place of work. Notably, there were very few internet connectivity issues during the Zoom interviews, with visuals and audio remaining largely stable throughout. Each interview was audio recorded and transcribed verbatim by a specialist transcription service. CH checked the transcripts to ensure accuracy.

## Data analysis

Following guidance [34] on using the TDF in qualitative studies and under the supervision of ED, CH developed coding guidelines ("a set of explicit statements of how the TDF is to be applied to a specific data set" [34].) Transcripts were imported into QSR NVivo 12 [35] for analysis.

A deductive approach was initially taken in which the researchers read the transcripts, considered the relevance of the data to the TDF's domains and theoretical constructs, and then coded the data into one or more of the 14 theoretical domains [32,34]. This was followed by thematically analysing [36] the data coded to each theoretical domain to generate belief statements. A belief statement is a "collection of responses with a similar underlying belief that suggest a problem and/or influence of the beliefs on the target implementation problem" [34]. In line with standard practice with TDF studies [34,37], once coding was complete, three criteria were considered when judging the relevance of the TDF domains and associated belief statements to the target behaviour: (1) a high frequency of coding ($\geq$80% participants), (2) presence of conflicting belief statements, and (3) presence of strong beliefs which may impact behaviour.

All transcripts were analysed by CH, while ED independently coded a subset of transcripts to check for consistency of coding. Differences in coding were discussed and the coding guidelines were iteratively revised until coding was acceptably consistent.

## Ethical considerations

The study was approved by The University of Melbourne Centre for Youth Mental Health Human Ethics Advisory Group (Ethics ID: 2056869). All participants were provided with written study information and signed a consent form prior to interview.

**Table 1. Participant characteristics.**

|  | Healthcare commissioners | YMH services |
|---|---|---|
| *No. of participants* | *20* | *20* |
| *No. (%) of commissioning regions represented from total of 31* | *18 (58.06%)* | *15 (48.36%)* |
| *Participant roles (No. of participants)* | • *Manager or program officer for youth mental health\* (17)*<br>• *Data or ME-related\* (3)* | • *Middle management (10)*<br>• *Clinical management (4)*<br>• *Data or ME-related (3)*<br>• *Project management (2)*<br>• *Clinician (1)* |
| *Types of YMH services represented (No. of participants)* | *NA* | • *headspace centres (11)*<br>• *Other commissioned YMH services (9)* |

*All healthcare commissioner participants had experience of working directly with YMH services.

## Results

### Participant overview

A total of 40 participants were recruited across both sample groups and data saturation of themes was achieved. The healthcare commissioners sample included staff responsible for the youth mental health portfolio, or staff involved in analysing YMH service data. The YMH services sample included management and other staff involved in ME from a commissioned YMH service.

Interviews lasted between 40 and 70 minutes (M = 56.63). Participant characteristics are summarised in Table 1.

### Current practice

The types of ME behaviours performed by healthcare commissioners and YMH services are shown in Table 2. Involvement in evaluation was mentioned by a few participants, but most ME activity related to monitoring only.

Although there are commonalities in the types of behaviours, there is variation in how these behaviours are performed by services and commissioning organisations. While all

**Table 2. ME behaviours performed by healthcare commissioners and YMH services.**

| ME behaviour | Healthcare commissioners | YMH services |
|---|---|---|
| *Planning and preparing for ME* | Y | Y |
| *Entering data into data systems* | N | Y |
| *Providing technical assistance to YMH services* | Y | N |
| *Retrieving data from data systems* | Y | Y |
| *Preparing monitoring reports for healthcare commissioners* | N | Y |
| *Analysing and visualising data* | Y | Y |
| *Providing feedback* | Y | Y |
| *Reviewing and interpreting data* | Y | Y |
| *Making decisions and taking action* | Y | Y |
| *Informal communication between healthcare commissioners and YMH services* | Y | Y |

healthcare commissioners require services to collect data, there are differences in the extent of these requirements. Some commissioners only require services to collect a nationally mandated primary mental health care minimum data set [38]. However, most commissioners require services to collect data in addition to this mandated data set and to provide monitoring reports (usually quarterly) which include quantitative data on service activities and outcomes, and qualitative data such as case studies or narrative. In addition to these formal monitoring mechanisms, many commissioners maintain informal communication with services to ensure they are up to date with what is happening and aware of any potential issues.

The degree to which commissioners and YMH services partner on ME varies. The ME planning process appears to be highly collaborative in some cases, while highly prescriptive in others. Similarly, some commissioners actively engage services in data-informed discussions (e.g. service development workshops), while other services report receiving little to no feedback on the reports they provide to their commissioners.

Who performs ME behaviours varies across services. For example, some services have specific data or ME-related staff who can retrieve data from data systems, and analyse and visualise data. However, in other services, staff may perform these behaviours on top of their formal job role (e.g. clinicians preparing commissioner reports). Services that operate as part of the headspace franchise are supported by the headspace National Office, which collects and analyses data from all centres, and provide centres and their commissioners with data reports and access to an online data visualisation tool.

## Domains analysis

Table 3 overviews which TDF domains were relevant for the behaviours. Twelve of the 14 domains were relevant to healthcare commissioner behaviours and 11 to YMH service behaviours. S2 Appendix provides detailed information regarding the frequency of TDF coding and belief statements, the rationale for relevance, and illustrative quotes.

## Belief statements shared by healthcare commissioners and YMH services

In total, 100 belief statements relating to healthcare commissioners and/or YMH service behaviours were generated. All belief statements, reasons for relevance, and illustrative quotes can be found in S2 Appendix. There were 26 belief statements that applied to both healthcare commissioner and YMH service behaviours, which are summarised in Table 4. Each of these belief statements are subsequently described in further detail (with the relevant TDF domains in bold), as well as those that were only held by one sample group.

Participants in both groups regarded ME as integral to their work (**intentions**). Many believed that ME should be primarily used to drive quality improvement (**intentions**) so that young people receive the best care possible and experience improved outcomes (**goals**). Numerous participants purported a desire to improve the use of ME (**intentions**) and improving ME planning was seen as key enabler of this (**behavioural regulation**). However, it was also widely acknowledged that ME is burdensome for services (**beliefs about consequences**) and that there are limited funds for them to allocate to it (**environmental context and resources**).

ME helps participants to understand what is happening in services, identifies service risks and gaps, informs service improvements, and guides healthcare commissioners on how they can support services (**beliefs about consequences**). Participants also had particular interest in using ME to understand the experiences of young people and families accessing services (**goals**). The inclusion of qualitative data in monitoring reports was regarded as essential by many, as it helps to contextualise quantitative data (**beliefs about consequences**).

**Table 3. TDF domains [32] and reasons for relevance/irrelevance.**

| Domain | Healthcare commissioners | | Youth mental health services | |
|---|---|---|---|---|
| | Relevant | Reasons for relevance/irrelevance | Relevant | Reasons for relevance/irrelevance |
| **Knowledge**<br>*An awareness of the existence of something* | N | *No evidence of strong beliefs that may impact on behaviour present* | N | *No evidence of strong beliefs that may impact behaviour present* |
| **Skills**<br>*An ability or proficiency acquired through practice* | Y | *High frequency, strong beliefs that may impact on behaviour, beliefs shared with YMH services* | Y | *High frequency, beliefs shared with healthcare commissioners, strong beliefs that may impact on behaviour* |
| **Memory, attention and decision processes**<br>*The ability to retain information, focus selectively on aspects of the environment and choose between two or more alternatives* | Y | *Strong beliefs that may impact on behaviour* | N | *Low frequency, no evidence of strong beliefs that may impact on behaviour* |
| **Behavioural regulation**<br>*Anything aimed at managing or changing objectively measured actions* | Y | *High frequency, beliefs shared with YMH services, strong beliefs that may impact behaviour* | Y | *High frequency, strong beliefs that may impact behaviour, beliefs shared with healthcare commissioners* |
| **Environmental context and resources**<br>*Any circumstance of a person's situation or environment that discourages or encourages the development of skills and abilities, independence, social competence, and adaptive behaviour* | Y | *High frequency, conflicting beliefs present, strong beliefs that may impact behaviour, beliefs shared with YMH services* | Y | *High frequency, conflicting beliefs present, strong beliefs that may impact behaviour, beliefs shared with healthcare commissioners* |
| **Social influences**<br>*Those interpersonal processes that can cause individuals to change their thoughts, feelings, or behaviours* | Y | *High frequency, strong beliefs that may impact behaviour, conflicting beliefs present, beliefs shared with YMH services* | Y | *High frequency, strong beliefs that may impact behaviour, conflicting beliefs present, beliefs shared with healthcare commissioners* |
| **Social professional role and identity**<br>*A coherent set of behaviours and displayed personal qualities of an individual in a social or work setting* | N | *Low frequency, no evidence of strong beliefs that may impact on behaviour* | N | *Low frequency, no evidence of strong beliefs that may impact behaviour* |
| **Beliefs about capabilities**<br>*Acceptance of the truth, reality, or validity about an ability, talent, or facility that a person can put to constructive use* | Y | *High frequency, strong beliefs that may impact behaviour* | Y | *High frequency, conflicting beliefs present* |
| **Emotion**<br>*A complex reaction pattern, involving experiential, behavioural and physiological elements, by which the individual attempts to deal with a personally significant matter or event* | N | *Low frequency, no evidence of strong beliefs that may impact behaviour* | Y | *Strong emotions present* |
| **Optimism**<br>*The confidence that things will happen for the best or that desired goals will be attained* | Y | *High frequency, demonstrated high level of optimism* | N | *No evidence of strong beliefs that may impact behaviour* |
| **Intentions**<br>*A conscious decision to perform a behaviour or a resolve to act in a certain way* | Y | *High frequency, strong beliefs that may impact behaviour, beliefs shared with YMH services* | Y | *High frequency, strong beliefs that may impact behaviour present, beliefs shared with healthcare commissioners* |
| **Beliefs about consequences**<br>*Acceptance of the truth, reality or validity about outcomes of a behaviour in a given situation* | Y | *High frequency, strong beliefs that may impact behaviour, conflicting beliefs present, beliefs shared with YMH services* | Y | *High frequency, conflicting beliefs present, beliefs shared with healthcare commissioners, strong beliefs that may impact behaviour* |
| **Goals**<br>*Mental representations of outcomes or end states that an individual wants to achieve* | Y | *High frequency, strong beliefs that may impact behaviour, conflicting beliefs present, beliefs shared with YMH services* | Y | *High frequency, beliefs shared with healthcare commissioners, strong beliefs that may impact behaviour* |
| **Reinforcement**<br>*Increasing the probability of a response by arranging a dependent relationship, or contingency, between the response and a given stimulus* | Y | *High frequency, strong beliefs that may impact behaviour, conflicting beliefs present* | Y | *High frequency, strong beliefs that may impact behaviour* |

"We receive monthly data and they've got a target and an achievement. Really they're only numbers on paper, until you understand what they actually mean. So I find that the qualitative stuff behind the data is of equal importance, because it speaks to the data. I think that tells us the richest information."

*(Participant 24, healthcare commissioner).*

**Table 4. Belief statements shared by healthcare commissioners and YMH services.**

| TDF domain | Belief statement |
| --- | --- |
| *Skills* | *You need to be able to build relationships with other organisations.* |
| | *You need a good understanding of the YMH service context.* |
| | *You need to be data literate.* |
| | *You need to be able to empathise with YMH service staff.* |
| | *You need to be inquisitive and open minded.* |
| *Behavioural regulation* | *Improvements in monitoring and evaluation planning.* |
| | *Improvements in data processes and tools.* |
| | *Greater access to data collected by headspace centres.* |
| *Environmental context and resources* | *I am able to access staff with monitoring and evaluation-related skills.* |
| | *I lack the time to dedicate to monitoring and evaluation.* |
| | *My organisation is supportive of the use of monitoring and evaluation information.* |
| | *Data processes and tools are problematic.* |
| | *Commissioned services have limited funds to allocate to monitoring and evaluation.* |
| | *It can be difficult to use the data we receive from headspace National Office for monitoring and evaluation.* |
| | *It feels like headspace centres have two masters.* |
| *Social influences* | *It's useful to access the support of national youth mental health organisations (e.g. Orygen and headspace National).* |
| *Intentions* | *Monitoring and evaluation is an integral part of the work we do.* |
| | *We intend on improving our use of monitoring and evaluation.* |
| *Beliefs about consequences* | *It helps me to understand what is happening in the service and informs improvement.* |
| | *It helps to identify service issues (including risks and gaps).* |
| | *The data does not always accurately reflect what's happening on the ground.* |
| | *Qualitative data is needed to contextualise quantitative data.* |
| | *Monitoring and evaluation is burdensome for services.* |
| *Goals* | *I want to understand the experiences of young people and families.* |
| | *Monitoring and evaluation helps to ensure young people receive the best care possible and experience improved outcomes.* |
| | *Monitoring and evaluation should inform quality improvement.* |

*"I think it's really the case studies that are particularly useful, because we can really get a good sense, ourselves, around what the presentations were for young people, what their goals were, what our evidence-based approaches were to meeting those goals, where the young person came to in their trajectory, and what the outcomes were for good, for bad, for otherwise, and also what the service impacts have been within service."*

*(Participant 6, YMH service).*

Being data literate, inquisitive and open minded were regarded as important ME skills by numerous participants (**skills**). Similarly, having a good understanding of the YMH service context (**skills**) was seen as important, as was being able to empathise with YMH service staff, and being able to build relationships with organisations (e.g. service providers, healthcare commissioning organisations) (**skills**). One healthcare commissioner reflected on value of having previously worked in a service:

"I understand the tensions within the work. Sure every service is different and I could never possibly say that I understand exactly what they're encountering on a day to day basis. . .

but as a general rule, having service delivery experience really does help you when you're collaborating with providers."

*(Participant 11, healthcare commissioner).*

Both groups acknowledged that ME data does not always accurately reflect what happens on the ground in services (**beliefs about consequences**). Several YMH service participants noted that reporting data from a single or limited number of outcome measures only provides a partial insight into the difference their service makes.

"I don't think that any of those measures should be taken individually. I think that would be reductionistic. . . they all need to be collected and viewed as a whole. I think to take any one of them individually and use that as the basis for the outcome is totally not valid."

*(Participant 8, YMH service).*

Commissioners provided a different perspective on this issue. One participant spoke of finding out that a service had withheld important information about challenges they were experiencing from the commissioner, while others spoke of the integrity of data sets being reduced by data entry issues within services. Problems with data processes and tools were also widely cited by both groups as barriers to ME (**environmental context and resources**).

*"We're just trying to enter things into multiple platforms and you do see differences in different platforms with even just caseloads and occasions of service numbers. They are slightly different and I think that's because we're trying to work across too many systems."*

*(Participant 28, YMH service).*

*"We have a database that the PHN [healthcare commissioning organisation] manages, which all of the service providers enter into. . . it does come with some challenges because the service providers often have a lot of difficulty—it's not the best system. It's quite limited in what it can do with reporting. So the service providers often have challenges in being able to export and being able to filter according to the KPIs. . ."*

*(Participant 3, healthcare commissioner).*

## Healthcare commissioner belief statements

The value of engaging with services on an ongoing, informal basis was raised by many (**beliefs about consequences**), and there was a strong desire among participants to develop stronger partnerships with services, so they can support them with quality improvement (**goals**). It was, however, mentioned that ME can identify service issues which the commissioner may not be able to help resolve (**beliefs about capabilities**).

*"I think the downside will probably be if I've found a need and I can't support that. . . So if I'm aware of a gap or if I'm aware that someone is struggling and I can't assist, I think that's sort of a negative of evaluation."*

*(Participant 34, healthcare commissioner).*

While some reported having little contact with other healthcare commissioning organisations regarding ME, many spoke of how they learn from and collaborate with other commissioners (**social influences**).

"I actually spoke to four other PHNs [healthcare commissioning organisations] to get their data to see what they collected and what some of their turnaround times were, which was fantastic. So we've done our own little benchmark study."

*(Participant 12, healthcare commissioner).*

The ability to develop appropriate expectations and performance indicators for commissioned YMH services was viewed as a vital skill by several participants (**skills**). One commissioner spoke of the dangers of setting inappropriate performance indicators:

*"I think people underestimate how hard it is to develop a really good indicator. . . You have to be really careful because you can create perverse incentives."*

*(Participant 16, healthcare commissioner).*

Some mentioned that the way in which the government measures healthcare commissioner performance incentivises a focus on service activity rather than service outcomes (**reinforcement**). Others spoke of how the national primary mental health care minimum data set is of limited use when monitoring and evaluating commissioned services (**environmental context and resources**).

"The PMHC-MDS [national primary mental health care minimum data set] is not fit for purpose. It has too many fields. It collects information that we don't necessarily use or value. It creates a reporting burden for provider organisations that's unnecessary and unwarranted."

*(Participant 17, healthcare commissioner).*

## YMH service belief statements

Several service participants indicated that doing ME helps to ensure their service retains funding from their commissioner because it is a contractual obligation (**reinforcement**), while others spoke of wanting to use ME to demonstrate the difference their service makes (**goals**). However, most participants indicated that ME often takes a backseat to other priorities, such as attending to the needs of young people and staff (**goals**).

Many felt that their commissioner actively supported them, but this feeling was not shared by all (**social influences**):

"The PHNs [healthcare commissioners] that I find helpful are the ones who are willing to work in partnership. . . there are commissioners who have described themselves as like an ATM: 'you complete the transaction and we give you the money'. Whereas others are more likely to work in partnership, so really collaborative kind of decision making."

*(Participant 31, YMH service).*

Numerous participants expressed that they felt their commissioner's expectations of their service was unrealistic (**reinforcement**). This related to either the volume of ME activity (i.e.

data collection, reporting) required of services or expectations about service performance. Some participants said they were worried about the potential consequences of not meeting the commissioner's expectations (**emotion**).

> *"It can also make me feel nervous. I guess I had a lot of anxiety when we'd had to do the Q3 report when I'd first started and I had to put zero next to a lot of our KPIs. That was very anxiety provoking."*
>
> *(Participant 10, YMH service).*

Participants suggested that commissioners could help YMH services with ME by collaborating with them (and young people and families) on decisions about ME planning, streamlining reporting requirements, and improving feedback (**behavioural regulation**).

It was widely reported that staff need to feel that data collection is meaningful for them to actively engage in it, and it was beneficial to create formal opportunities to discuss data with staff (**behavioural regulation**).

While participants asserted that ME should benefit clinical practice (**goals**), there were mixed views about its impact (**beliefs about consequences**), particularly in regard to using outcome measures with young people. While several participants spoke about the value of using measures, some felt that using measures that focus on symptoms and problems can inhibit recovery-orientated practice. Participants also spoke of how clinicians value the use of data in their practice to varying degrees (**social influences**). For many, the use of ME helps to ensure that their service operates in an evidence-based way (**beliefs about consequences**).

> "Without evaluation and reflection and looking at ourselves and looking at what we're doing, we could be in the dark ages. We could be providing a service that is unhelpful. . . Evaluation means that we can't not be focused on outcomes in the participant and their needs, and keeps us ethical, and keeps us up-to-date with best practice."
>
> *(Participant 33, YMH service).*

## Discussion

This study sought first to explore how data collection and feedback practice, in the form of monitoring and evaluation (ME), is used by YMH services and the organisations that commission them. Secondly, the study aimed to identify the barriers to and enablers of ME use from the perspectives of both YMH services and healthcare commissioners. We found that ME is a complex set of behaviours (e.g. planning and preparing for ME; entering data into data systems; providing technical assistance to YMH services; retrieving data from data systems; preparing reports for healthcare commissioners; analysing and visualising data; providing feedback; reviewing and interpreting data; making decisions and taking action; and informal communication between healthcare commissioners and YMH services). While there were commonalities in the types of behaviours performed, there was variation in how they were performed by commissioning organisations and YMH services. Both groups identified numerous individual, social, and environmental barriers and enablers. Many of these have the potential to be modified to enhance the use of ME activity to better support improving quality of service provision.

It was important for both commissioners and YMH services that data should drive service quality improvement. However, both groups raised concerns that data does not provide a fully accurate picture of service performance, and YMH services also felt that commissioners'

expectations of service performance were sometimes unrealistic or not meaningful. Difficulties with measuring quality in mental health care have been raised in previous literature [7,9,10,39]. In one study, mental health service managers reported that because performance indicators set for them did not obviously relate to service performance, data collection was regarded as a compliance activity rather than an opportunity to identify potential service improvements [39]. Beliefs articulated by participants in the present study can also be related to Mannion and Braithwaite's [40] taxonomy of dysfunctional outcomes of health performance measurement. The authors identify 21 unintended or adverse consequences relating to poor measurement, misplaced incentives or sanctions, breach of trust, and politicisation of performance systems [40].

Young people having a positive experience of care was of the utmost importance to commissioners and services alike. Many thus believed that data should provide meaningful insights that support them to improve patient experiences and outcomes. Literature suggests that providing clinicians with actionable feedback that presents aspects of care delivery that are under their control and relevant to their job has the greatest chance of making a difference to practice [41,42]. Yet, in this context, data collection focused primarily at the patient-level without a strong focus on clinician-level activities that contribute to patient outcomes. A greater focus on clinician-level data in ME plans may help to ensure that data optimally contributes to improving the experiences of young people receiving care.

While clinician-level data is critical to actionable quality improvement, patient-level data is also important in measuring quality of care and clinical decision-making [2,6,7,13]. The role of outcome measurement in this was a topic of contention in this study. Participants reported variability in the value that clinicians place on using measurement in their practice and regarded mandated outcome measures to be of limited clinical utility or even a potential impediment to recovery-orientated practice. These issues are consistent with the literature on implementing outcome measurement in mental health settings [6,43–48]. To avoid the risk of it becoming a purely bureaucratic exercise, outcome measurement should be meaningful to clinicians, young people, and families and carers [6,45,49–52]. The dearth of clinically meaningful outcome measures designed for young people has been previously highlighted [53], but such measures are being developed [54,55]. It has also been advocated that using idiographic outcome measures (e.g. Goal-Based Outcome Tool) [56] can help to facilitate person-centred care [51,57,58] and has been associated with improvements in young people's satisfaction and engagement with mental health services [59,60].

Challenges relating to data processes and systems, and minimum data sets are well documented in the literature [7,9,39,43,61,62], and consistent with the results of the present study. It was common for participants to speak of the burden of having to use multiple systems because of a lack of interoperability between systems or because data were needed that was not available in the national primary mental health care minimum data set. It is a priority for commissioners that YMH services collect the minimum data set because that data is used by the Australian government to measure commissioner performance, but for many commissioners, the minimum data set does not adequately capture service performance. This places commissioners in a challenging position. They are mindful that ME places a significant burden on YMH services but it is difficult for them to meaningfully monitor and evaluate services without requiring the collection of additional data.

Lastly, commissioners and YMH services want to work in partnership and such an approach may help to address some of the challenges. Services spoke of the benefits of commissioners being collaborative and forthcoming with meaningful feedback and commissioners spoke of how valuable they found it to communicate with services on an ongoing and informal basis ('soft governance'), which aligns with the commissioning literature [62–64]. Both groups

also regarded interpersonal skills such as the abilities to empathise and build relationships as essential. This corroborates existing research, which emphasises that a trusting relationship between the provider and recipient of feedback improves the likelihood that the feedback will inform learning and improvement [65,66].

## Strengths and limitations

The inclusion of both YMH service and healthcare commissioner perspectives from a good spread of geographical regions is a strength of the study. A limitation, however, is that a significant number of participants were recruited through the researcher's professional network, so there is a potential risk of self-selection bias. Given the significance that participants placed on understanding the experiences of young people and families, future research is needed to include their views in ME activity.

The use of the Theoretical Domains Framework also has strengths and limitations. The TDF's 14 domains, underpinned by 33 behavioural theories, enabled the identification of a wide range of barriers and enablers to ME. Systematically exploring each of the TDF's domains in the interviews may have helped unveil barriers and enablers that would have been otherwise missed. It also allows for the results of this study to be used in the development of strategies to enhance ME use, through mapping the relevant TDF domains to behaviour change theory (i.e. the behaviour change wheel approach to intervention design) [34,67]. However, it would be valuable to critically appraise the data when designing these strategies, as prior research shows that people tend to emphasise external (environment, social influences) rather than internal (knowledge, skills) factors as barriers to their own behaviour [68,69]. Finally, while the TDF is extensive in its scope, it is possible that there are barriers and enablers that are not currently covered by its 14 domains.

## Implications for practice

There are several strategies emerging from this research that healthcare commissioners and YMH services should implement to ensure ME is meaningful.

Firstly, ME plans should be co-designed [70] and should meaningfully involve young people and families whenever possible. The targets, performance indicators and outcome measures should explicitly link to YMH service quality improvement and, where possible, provide clear examples to demonstrate how improvements can be achieved. ME plans should also include qualitative data such as case studies.

Streamlined data collection processes will reduce unnecessary burden, and YMH services should have the capability to interrogate their own data and generate reports. Healthcare commissioners should also ensure that they provide meaningful feedback to their commissioned services, and local and national organisations collecting youth mental health data should facilitate the sharing of this data.

YMH services and commissioners should be provided with opportunities to build their ME capacity by organisations with relevant expertise. Finally, it must be noted that additional investment will likely be required for YMH services and commissioners to implement these recommendations.

## Implications for future research

Future research could identify what targets, performance indicators and outcome measures would be most appropriate to use in youth mental healthcare. Young people and families should be meaningfully involved in this research, particularly in the development and ongoing validation of outcome measures.

## Conclusions

By using a theory-informed behavioural approach to explore the use of ME in youth mental health care we found that current practice comprises of numerous interrelated behaviours performed by YMH services and healthcare commissioners, and that there are many barriers and enablers to this activity at the individual; organisational; and broader environmental levels. Importantly, this study illustrated scope for improvement. The results of the study should be used to design theory-informed strategies to improve ME use. This would help to ensure that the use of ME produces 'more than just numbers on a page' and leads to continuous improvements in the quality of mental health care provided to young people.

## Supporting information

**S1 Appendix. Interview topic guide.**
(PDF)

**S2 Appendix. Domains, beliefs, rationale table.**
(PDF)

## Acknowledgments

We would like to thank the participants who generously shared their time and experience for the purposes of this study.

## Author Contributions

**Conceptualization:** Craig Hamilton, Eilidh Duncan.

**Formal analysis:** Craig Hamilton, Eilidh Duncan.

**Investigation:** Craig Hamilton.

**Methodology:** Eilidh Duncan.

**Project administration:** Craig Hamilton, Kate Filia.

**Supervision:** Eilidh Duncan.

**Writing – original draft:** Craig Hamilton.

**Writing – review & editing:** Craig Hamilton, Kate Filia, Sian Lloyd, Sophie Prober, Eilidh Duncan.

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
