## [Decision Letter · Decision Letter 0]

1 Apr 2022

PONE-D-22-06227‘More than just numbers on a page?’ A qualitative exploration of the use of data collection and feedback in youth mental health services.PLOS ONE

Dear Dr. Hamilton,

Thank you for submitting your manuscript to PLOS ONE. After careful consideration, we feel that it has merit but does not fully meet PLOS ONE’s publication criteria as it currently stands. Therefore, we invite you to submit a revised version of the manuscript that addresses the points raised during the review process.

We look forward to receiving your revised manuscript.

Kind regards,

Jason Scott

Academic Editor

PLOS ONE

Journal Requirements:

Reviewers' comments:

Reviewer's Responses to Questions

**Comments to the Author**

1. Is the manuscript technically sound, and do the data support the conclusions?

Reviewer #1: Yes

Reviewer #2: Partly

2. Has the statistical analysis been performed appropriately and rigorously? 

Reviewer #1: N/A

Reviewer #2: No

3. Have the authors made all data underlying the findings in their manuscript fully available?

Reviewer #1: Yes

Reviewer #2: Yes

4. Is the manuscript presented in an intelligible fashion and written in standard English?

Reviewer #1: Yes

Reviewer #2: Yes

5. Review Comments to the Author

Reviewer #1: This is an interesting and well written manuscript on a topic that is important to ensure the sustainability of services. The conclusions drawn are appropriate based on the data presented and appropriate recommendations have been made. I thought the results were particularly will presented, highlighting the domains of the TDF. Throughout the manuscript authors should make explicit which behaviour is being performed and by whom.

Abstract

1. Design – ‘data was’ should be ‘date were’. Please check the full manuscript for this error.

2. Results – it’s unclear what ‘behaviour’ refers to here and what the ten types of behaviour are. How does this link to TDF?

3. Conclusions – Could the conclusions be better presented than a list? Makes it difficult to get a sense of impact.

Introduction

4. Good and concise overview of the evaluation and feedback.

5. It is highlighted there are two main gaps 1) to understand more about how data is used, and 2) barriers/enablers for improvement based on data collected. But the aims for the manuscript do not align with these and instead look to give an account of how data is collected and the barriers to collecting data from these services. Could authors please update this.

Methods

6. Within the study context – could authors please state what type of data is collected (clinical scores, wellbeing, engagement, etc.)?

7. The authors briefly discuss the use of zoom for most interviews. Could you please reflect on how this influenced data collection, e.g., were there any distractions for participants (such as shared office) that may have influenced answers? See for instance the below papers that examined benefits and drawbacks of Zoom:

Oliffe, J. L., Kelly, M. T., Gonzalez Montaner, G., & Yu Ko, W. F. (2021). Zoom interviews: benefits and concessions. International Journal of Qualitative Methods, 20, 16094069211053522.

Archibald, M. M., Ambagtsheer, R. C., Casey, M. G., & Lawless, M. (2019). Using zoom videoconferencing for qualitative data collection: perceptions and experiences of researchers and participants. International journal of qualitative methods, 18, 1609406919874596.

8. Could the authors please discuss whether they reached saturation / sufficient information power? e.g., whether they felt that saturation had been reached, they had reached sufficient information power, or whether they were time constrained?

Results

9. Throughout the results section could authors please state the number of participants rather than using ‘few’, ‘many’, ‘some’, ‘most’, etc.

10. Could authors please make clear what the 27 shared belief statements are- either through a clear description or use of a table/figure for example.

11. There are only nine supporting quotes presented intext and while these are representative there is more scope to include additional data to provide further meaning behind each belief.

Discussion

12. Authors mention ‘a complex set of behaviours’ performed. Could authors please make explicitly clear which behaviours are being referred to.

13. Authors reflect on their use of the TDF across the strengths and limitations, implications for future research, and conclusions sections. This makes the discussion of the TDF feel fragmented and would be best to have a central discussion within in the main section. The TDF highlights important conclusions and could be discussed in greater depth.

Appendix

14. Could authors please check appendix 2 table to ensure the illustrative quotes are representative of the specific beliefs. For example, Skills domain – Specific belief: You need to be data literate. I do not feel the quote is representative and instead reflects collaboration and belief about capability.

15. Could authors please check for syntax errors within the table (use of quotation marks, indenting text, missing letters from beginning of words).

Reviewer #2: This manuscript presents the results of a study of a qualitative study of beliefs about monitoring and evaluation (ME) from mental health commissioners’ and staff associated with youth mental health services in Australia. Forty separate 40-70-minute semi-structured interviews were conducted with participants recruited from a pool of 240+ potential participants using the authors’ social network and a snowball approach (asking first-level contacts to forward the invitation to others). A strength of the study is the use of a theory-based approach to developing the qualitative coding scheme using the Theoretical Domains Framework (TDF) to organize the a priori categorization of themes arising from the interviews. Study findings have the potential to further our understanding of ME in youth mental health services both in terms of use and implementation. However, there were several significant limitations that would need to be addressed to improve the scientific rigor and allow readers to better assess the implications of the findings.

First, the introduction mixes very separate conceptual areas by combining into a single construct the notions of routine outcome monitoring, audit and feedback, and monitoring and evaluation. While there is some overlap amongst these strategies for informing patient care and quality improvement efforts, they are different. Routine outcome monitoring, for example, has a single reference for a 2019 systematic review (6). There are dozens of studies and several systemic reviews and meta-analyses on this evidence-based practice available (e.g., de Jong et al., 2021, see reference below). The utility of this manuscript could be much clearer by focusing on ME and operationalizing what it is, how it is used, and what the known barriers are in the introduction.

Second, the methods description is not sufficiently rigorous for a qualitative study of high quality. For example, it is difficult to assess the quality of the recruitment process from what is described here. Was a single email sent? How does this differ from a sample of convenience? How were additional recruits vetted from the snowball sampling? In addition, a serious limitation is the lack of description of qualitative methods outside of the TDF coding scheme. What is meant by inductive and deductive coding and how did this arise from pre-planning coding schemes versus being developed as themes arose? How was quality maintained on the coding itself (e.g., common methods are consensus coding, use of training for inter-rater reliability, etc.). This reviewer is unclear what was meant by the three criteria described in paragraph 3 on page 6 – why these criteria, how and when were they applied, etc. Without further description, it is impossible for this reviewer to assess the quality of the study, nor to assess the interpretation of findings. In other words, there is little to no description of the analytic approach and methods.

de Jong, K., Conijn, J. M., Gallagher, R. A. V., Reshetnikova, A. S., Heij, M., & Lutz, M. C. (2021). Using progress feedback to improve outcomes and reduce drop-out, treatment duration, and deterioration: A multilevel meta-analysis. Clinical Psychology Review (85), 102002. https://doi.org/10.1016/j.cpr.2021.102002

6. PLOS authors have the option to publish the peer review history of their article (what does this mean?). If published, this will include your full peer review and any attached files.

Reviewer #1: **Yes: **Matthew Cooper

Reviewer #2: No

---

## [Author Response · Author response to Decision Letter 0]

15 May 2022

We thank reviewer 1 and 2 for taking the time to review the manuscript and the constructive feedback they have provided. We believe addressing them will strengthen the paper. We have detailed how we have addressed each piece of feedback in the attached document 'response to reviewers'.

---

## [Decision Letter · Decision Letter 1]

22 Jun 2022

‘More than just numbers on a page?’ A qualitative exploration of the use of data collection and feedback in youth mental health services.

PONE-D-22-06227R1

Dear Dr. Hamilton,

We’re pleased to inform you that your manuscript has been judged scientifically suitable for publication and will be formally accepted for publication once it meets all outstanding technical requirements.

Kind regards,

Jason Scott

Academic Editor

PLOS ONE

Additional Editor Comments (optional):

Reviewers' comments:

Reviewer's Responses to Questions

**Comments to the Author**

1. If the authors have adequately addressed your comments raised in a previous round of review and you feel that this manuscript is now acceptable for publication, you may indicate that here to bypass the “Comments to the Author” section, enter your conflict of interest statement in the “Confidential to Editor” section, and submit your "Accept" recommendation.

Reviewer #1: All comments have been addressed

Reviewer #2: All comments have been addressed

2. Is the manuscript technically sound, and do the data support the conclusions?

Reviewer #1: Yes

Reviewer #2: Yes

3. Has the statistical analysis been performed appropriately and rigorously? 

Reviewer #1: N/A

Reviewer #2: Yes

4. Have the authors made all data underlying the findings in their manuscript fully available?

Reviewer #1: Yes

Reviewer #2: Yes

5. Is the manuscript presented in an intelligible fashion and written in standard English?

Reviewer #1: Yes

Reviewer #2: Yes

6. Review Comments to the Author

Reviewer #1: Clear and easily understood. Thank you for addressing comments and presenting this insightful work.

Reviewer #2: With the new tables and figures, expanded detail in the methods section, and the clarification of focus on ME and not routine outcome monitoring (ROM) nor audit and feedback, the authors have thoroughly addressed this reviewer's comments. If interested, it is recommended that the authors update the seminal reference for ROM to the most recent meta-analysis (reference provided again below) rather than just a 2019 systematic review.

de Jong, K., Conijn, J. M., Gallagher, R. A. V., Reshetnikova, A. S., Heij, M., & Lutz, M. C. (2021). Using progress feedback to improve outcomes and reduce drop-out, treatment duration, and deterioration: A multilevel meta-analysis. Clinical Psychology Review (85), 102002. https://doi.org/10.1016/j.cpr.2021.102002

7. PLOS authors have the option to publish the peer review history of their article (what does this mean?). If published, this will include your full peer review and any attached files.

Reviewer #1: **Yes: **Matthew Cooper

Reviewer #2: **Yes: **Susan Douglas

---

## [Editor Report · Acceptance letter]

27 Jun 2022

PONE-D-22-06227R1 

‘More than just numbers on a page?’ A qualitative exploration of the use of data collection and feedback in youth mental health services. 

Dear Dr. Hamilton:

I'm pleased to inform you that your manuscript has been deemed suitable for publication in PLOS ONE. Congratulations! Your manuscript is now with our production department. 

Kind regards, 

on behalf of

Dr. Jason Scott 

Academic Editor

PLOS ONE